# Forecasting unprecedented ecological fluctuations

**Samuel R. Bray, Bo Wang***

Department of Bioengineering, Stanford University, Stanford, California, United States of America

* wangbo@stanford.edu

## Abstract

Forecasting 'Black Swan' events in ecosystems is an important but challenging task. Many ecosystems display aperiodic fluctuations in species abundance spanning orders of magnitude in scale, which have vast environmental and economic impact. Empirical evidence and theoretical analyses suggest that these dynamics are in a regime where system nonlinearities limit accurate forecasting of unprecedented events due to poor extrapolation of historical data to unsampled states. Leveraging increasingly available long-term high-frequency ecological tracking data, we analyze multiple natural and experimental ecosystems (marine plankton, intertidal mollusks, and deciduous forest), and recover hidden linearity embedded in universal 'scaling laws' of species dynamics. We then develop a method using these scaling laws to reduce data dependence in ecological forecasting and accurately predict extreme events beyond the span of historical observations in diverse ecosystems.

**Data Availability Statement:** Data for the plankton community can be accessed from the supplementary section of the paper [Benincà E. et al, "Coupled predator-prey oscillations in a chaotic food web" (2009)]. Data for the intertidal communities can be accessed from the

## Author summary

Rare large-amplitude 'Black Swan' fluctuation events have significant ecological and economic impact. In this work, we tackle the grand challenge in forecasting critical fluctuations in ecosystems, in particular in data sparse regimes. We take an unconventional approach by bridging the fields of statistical physics and ecological forecasting. We apply theory from avalanche systems (such as earthquakes) to analyze long-term monitoring data from diverse natural ecosystems, including marine plankton, intertidal mollusks, and deciduous forest. These datasets allow us to recover the clean power-law relations, or 'scaling laws' in statistical physics terms, in system fluctuations that are ubiquitous across species and communities. Leveraging these scaling laws, we extrapolate rare, extreme dynamics from limited historical data and accurately forecast unprecedented events. Therefore, our results have the potential to maximize data value in ecological forecasting with a broad set of applications.

## Introduction

Theoretical models and long-term tracking of natural and experimental ecosystems have identified widespread aperiodic fluctuations spanning orders of magnitude in amplitude and

supplementary section of the paper [Benincà E, et al. "Species fluctuations sustained by a cyclic succession at the edge of chaos" (2015)]. Data for the Harvard forest can be accessed at: http://dx.doi.org/10.17190/AMF/1246059 and is reported in [Urbanski S, et al. "Factors controlling $CO_2$ exchange on timescales from hourly to decadal at Harvard Forest." (2007)]. Code for the analysis and plotting of data is available at the Github repository: https://github.com/samuelbray32/ecologicalAvalanches.

**Funding:** This work is funded by Volkswagen Foundation (No. 94819). SRB is supported by a NIH CMB training grant (T32GM007276). BW is supported by a Beckman Young Investigator Award. The funders had no role in study design, data collection and analysis, decision to publish, or preparation of the manuscript.

**Competing interests:** The authors have declared that no competing interests exist.

duration [1–7]. These fluctuations, especially rare but large-amplitude 'Black Swan' events, have significant ecological and economic impact [7,8], generating a pressing need for forecasting and management strategies. However, the highly nonlinear, near-chaotic nature of these fluctuations presents significant challenges for forecasting algorithms [4,6,7,9].

Ecological forecasting methods tend to fall into two categories. The first uses prior biological knowledge and statistical inference of historical data to build and fit a parametric model of the ecosystem [7,10–12]. This requires extensive information for each community under study. These models are also prone to overfitting to the current system conditions, with poor generalization to rare and extreme events. The second approach, empirical dynamic modeling (EDM), relies on reconstructing attractors of system dynamics [6,13,14]. Single- or multi- species data are embedded into a high-dimensional state space. Forecasts from the current time point are made based on the trajectories of neighboring historical states. These methods can account for species interactions that vary nonlinearly with the system state and have proven useful in predicting ecological dynamics. However, they can suffer from poor extrapolation during rare events when no closely neighboring trajectories are present in historical data. These methods are all fundamentally limited by the information within historical data, which often fails to contain all possible dynamics of a system due to the costs of long-term monitoring and low frequency of extreme events [15,16]. Therefore, we sought a method to better extrapolate the dynamics of large, rare events from historical data containing only small but frequent fluctuations.

To achieve this goal, we turn to the concept of 'avalanches' from statistical physics. Avalanches are a specific type of discrete, impulsive fluctuation characterized by heavy-tailed power law distributions (or scaling laws) in size and a self-similarity between events of divergent scales that can arise from various microscopic generative processes [17,18]. These dynamics occur in many systems including earthquakes, magnet polarization, and neuronal firings [17–19]. Despite the diversity of systems that produce avalanches, the resulting dynamic scaling laws often fall into a limited set of so-called "universality classes" that are insensitive to the system specific details [17,20]. We reason that such scaling laws will allow us to use the information inherent to the universal features of fluctuations to augment that of limited historical data and improve forecasting accuracy.

Previous work has shown indications of avalanche-like dynamics in ecosystems using measures of species abundance and persistence [21–27], but due to data availability, these studies typically considered a single distribution aggregated across species with timescale resolution on the order of species' lifetimes. Our fluctuation forecasting method goes beyond these demonstrated relations and utilizes the full information about fluctuation duration, size, and frequency within a single species lifetime. Validating these relations requires extensive data with both high sampling frequency (to sample short events) and long duration (to capture rare events) across multiple diverse communities (to demonstrate the conservation of the scaling laws). Here we utilize the recently accumulating body of long term ecological research and analyze the dynamics of three ecosystems: Baltic Sea plankton abundance measured in a reconstructed closed laboratory [4,28], mollusk coverage in intertidal zones at Goat Island Bay, New Zealand [3], and $CO_2$ fluxes in the Harvard deciduous forest [29]. All three systems have decade(s) of tracking data with sampling frequency varying from minutes to weeks. Each of the three systems shows persistent large-amplitude bursts in the absence of extreme environmental events in a manner consistent with avalanche dynamics (**Fig 1A**). We first validate a complete set of self-consistent scaling laws conserved across communities and independent of species interactions. We then develop a method based on the scaling laws to accurately extrapolate historical data to unsampled regimes and improve forecasting during unprecedented events.

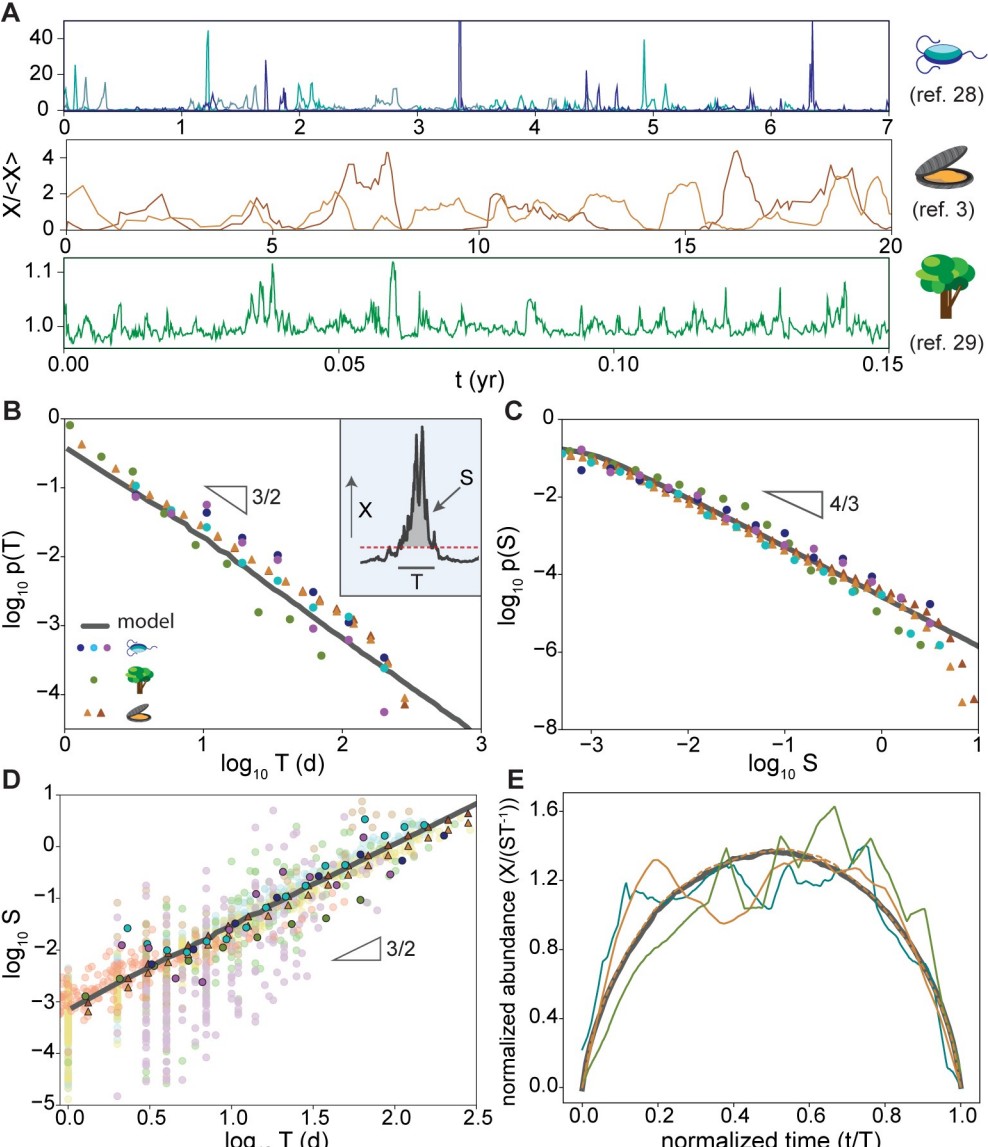

**Fig 1. Ecological fluctuations show universal avalanche scaling laws. A,** Time series of species abundance (protozoa, rotifers, calanoids) in an isolated plankton community [28] (top), rock coverage by intertidal mollusks [3] (middle), and forest carbon fluxes [29] (bottom) show sustained aperiodic fluctuations varying in size and duration. Magnitudes (X) are normalized by mean value for each dataset. **B,** The probability distributions of avalanche durations follow a universal power law across communities. Avalanche duration (T) and corresponding size (S) are defined in **inset**. Blue, cyan, purple: herbivore, detritivore, and producer trophic levels in the plankton community; green: forest carbon flux; brown, yellow: barnacle and algae in the intertidal community. Due to the small number of events in the intertidal community, this dataset is supplemented with the model provided in the original paper (triangle) [3]. The empirical data are in quantitative agreement with the linear response model at marginal stability (**Eq 1**, $\lambda = 10^{-3}$, grey line). **C,** The distributions of avalanche sizes follow a universal power law. Sizes are shifted horizontally to account for varied units between datasets. **D,** Scaling of average avalanche size vs. duration follows a power-law conserved across communities and the linear response model (grey line). Translucent symbols: individual avalanche events; solid symbols: bin-averaged values. In **B-D**, individual groups are collapsed through leading coefficients. **E,** Average avalanche shape converges across datasets and the linear response model. Each avalanche event is isolated and normalized to unit duration ($t' = t/T$) and unit size ($X' = X/ST^{-1}$, resulting in $S' = \int_0^1 X'(t')dt' = 1$). Trajectories from each dataset are then averaged to produce the avalanche shape separately. Blue, yellow, green: plankton, intertidal, and forest data. Dashed line: modeled intertidal community. Grey line: linear response model at marginal stability ($\lambda = 10^{-3}$).

## Results

### Fluctuation scalings across natural communities

Avalanche dynamics are defined by a specific set of scaling relations between fluctuation duration, frequency, and size across broad scales [20,26,30]. Previous theoretical analyses have suggested that these scaling laws may experience a crossover at long observational timescales in ecological fluctuations [30], which may limit the applicability of these scaling relations in forecasting unprecedented large events. Therefore, we first sought to confirm all predicted scaling behaviors on relevant forecasting timescales.

To verify these scalings, we isolated avalanche events for each species as periods when data amplitude (X) exceeds a reference baseline level [17,31]. Each event has a duration (T) measured as the time between two intersections with the reference, size (S) calculated from the area of the data above the reference, and shape as the trajectory between two intersections (**Fig 1B**, inset). We found that all three datasets exhibit universal scaling laws regardless of where the reference baseline is set (**S1 Fig**)**,** with exponents matching theoretical expectations [17,20,30].

Specifically, the frequency of avalanche events of a given duration follows a power law across several decades with an exponent of $\alpha \sim 3/2$ (**Fig 1B**). The frequency of a given avalanche size follows another power law, with an exponent of $\tau \sim 4/3$ (**Fig 1C**). We verified the power-law and critical exponents of both distributions using the Akaike information content of maximum likelihood estimate (**Table 1**) to exclude potential fitting biases [32,33]. These heavy-tailed power-law distributions suggest that the size of the largest fluctuations should grow with observation time, explaining the previously documented difficulties in using finite data to sample the state space of an ecosystem [15].

Average avalanche size follows another power-law scaling against duration with an exponent of $\gamma \sim 3/2$ (**Fig 1D**, and **S1 Table**). This set of exponents, $\alpha$, $\tau$, and $\gamma$, gives a self-consistent characterization of the fluctuation dynamics, as confirmed by the relation $\gamma = \frac{\alpha-1}{\tau-1}$ [20]. While individual avalanches exhibit large variations, the scaling law between avalanche size and duration indicates that the average trajectory for avalanches of a given duration should be conserved across systems and time scales. To extract the average avalanche trajectory, we

**Table 1. Statistics of avalanche probability distributions.**

| Group | n | $\alpha$ MLE±95% CI | $\alpha$ AIC$_{max}$ | $\alpha$ AIC$_{theory}$ | $\tau$ MLE±95% CI | $\tau$ AIC$_{max}$ | $\tau$ AIC$_{theory}$ |
|---|---|---|---|---|---|---|---|
| Harvard forest | 2570 | 1.53±0.02 | 1.0 | 1.0 | 1.25±0.01 | 1.0 | 1.0 |
| Algae | >10$^5$ | 1.52±0.002 | 1.0 | 1.0 | 1.36±0.002 | 1.0 | 1.0 |
| Mussel | >10$^5$ | 1.49±0.002 | 1.0 | 1.0 | 1.34±0.002 | 1.0 | 1.0 |
| Herbivorous plankton | 148 | 1.59±0.14 | 1.0 | 1.0 | 1.38±0.08 | 1.0 | 1.0 |
| Photosynthetic plankton | 198 | 1.75±0.15 | 1.0 | 0.0007 | 1.36±0.07 | 1.0 | 1.0 |
| Detritivore | 121 | 1.82±0.16 | 1.0 | 1.0 | 1.46±0.08 | 1.0 | 1.0 |

To verify duration-frequency scaling ($\alpha$), the total number of avalanches (n) segmented within each group are fit to a power law, $p(T) = aT^{-\alpha}$, or an exponential distribution, $p(T) = a^{-\zeta T}$, through Maximum Likelihood Estimation (MLE) [32]. To avoid skewed results from the distribution cut-off, MLE fits were made with support on the range [0.95t$_s$, 100t$_s$] where t$_s$ is the sampling frequency of each system. The likelihood of each distribution is compared using the Akaike Information Criterion (AIC) to determine the weight of evidence supporting each distribution. Values of 1.0 given in cases where >0.9999 of the evidence is in support of the power law distribution. This analysis was repeated for both the MLE value $\hat{\alpha} = \hat{\alpha}_{max}$ (AIC$_{max}$) or $\hat{\alpha} = 3/2$ (AIC$_{theory}$), which is the predicted slope of the avalanche theory [20]. Similar analysis was performed for size-frequency scaling ($\tau$), with fits compared for the functions $p(S) = aS^{-\tau}$ and $p(S) = a^{-\zeta S}$. AIC$_{theory}$ was determined using $\hat{\tau} = 4/3$. Algae and mussel distribution data are from simulations using the mechanistic model described in the original study because experimental data have too few data points.

normalized each avalanche event by its mean amplitude ($ST^{-1}$) and duration ($T$) and computed the average normalized abundance at each time point along the trajectory (**Methods**). The results confirm our expectation that the average avalanche trajectory converges to a conserved shape across communities, though the averaging of noise is limited by the number of events in the experimental datasets (**Fig 1E**). The scaling laws and conserved avalanche trajectory are important as the linear relationship in log-log space provides a possibility to extrapolate the expected trajectory of large fluctuations from small events through nonlinear relationships.

## Avalanche scaling is consistent with marginal stability in near-neutral ecological dynamics

The conserved avalanche scaling suggests that these fluctuations are driven by a generic feature rather than system dependent species interactions. We tested whether the dynamics of near-neutral ecology are sufficient to reproduce the avalanche statistics. Near-neutral dynamics can emerge in systems with weak cross-species interactions compared to stochastic driving forces or the influences of environmental factors, and are often taken as a null model for ecological dynamics [30]. In this regime, the restoring force to perturbations becomes relatively weak, and stochasticity can drive extreme fluctuations, giving rise to a state of so-called "marginal stability" (**S2 Fig**) [34–36]. Empirical evidence of marginal stability has been found in ecosystems at the 'edge of chaos' [1–7]. However, the specific generative mechanism is expected to be irrelevant to statistical features of dynamic fluctuations and thereby the determination of scaling laws.

We use a linear response model to simulate dynamics near marginal stability (**Methods**). This model has been shown to generate signatures of emergent neutral ecology from large numbers of weak species interactions without the need to assume a specific interaction matrix [34–36]. Numerical simulations of the model produce avalanche scaling laws across increasingly wider dynamic ranges as the system approaches marginal stability (**S2 Fig**), with exponents in quantitative agreement with the empirical data (grey lines in **Fig 1B–1D**). While we acknowledge that this is not the only model that can generate neutral ecology behavior, it provides a well-converged form of average avalanche trajectory consistent with the data (**Fig 1E**). This allows us to use the model for generating test data in characterizing the performance of forecasting algorithms described below.

## Data extrapolation based on avalanche scaling improves forecasting during unprecedented fluctuations

We next applied the avalanche statistics to improve forecasting accuracy during ecological fluctuations. Incorporating this information into forecasting algorithms (particularly EDM methods) is non-trivial, as their predictions rely directly on time series of species abundance. Therefore, we developed a method, avalanche scaling extrapolation (ASE), to extrapolate dynamic trajectories of large, unprecedented events from available small, frequent fluctuations using the scaling laws identified above (**Fig 2A**, fitting sector). ASE calculates a single system-specific parameter (unit avalanche size) based on observed fluctuations. This is then combined with the scaling laws to stretch the average avalanche trajectory and produce expected dynamics of large fluctuations that are not present in the historical data (**Methods**). The generated time series data (referred to as ASE data for the rest of the paper) can be fed directly into EDM methods such as S-map [14]. In S-map, forecasts are made by estimating the linearized local dynamics based on neighboring historical data points in a time-embedded state space. The predictions are calculated via a linear regression of the trajectories of k-nearest neighbors, with

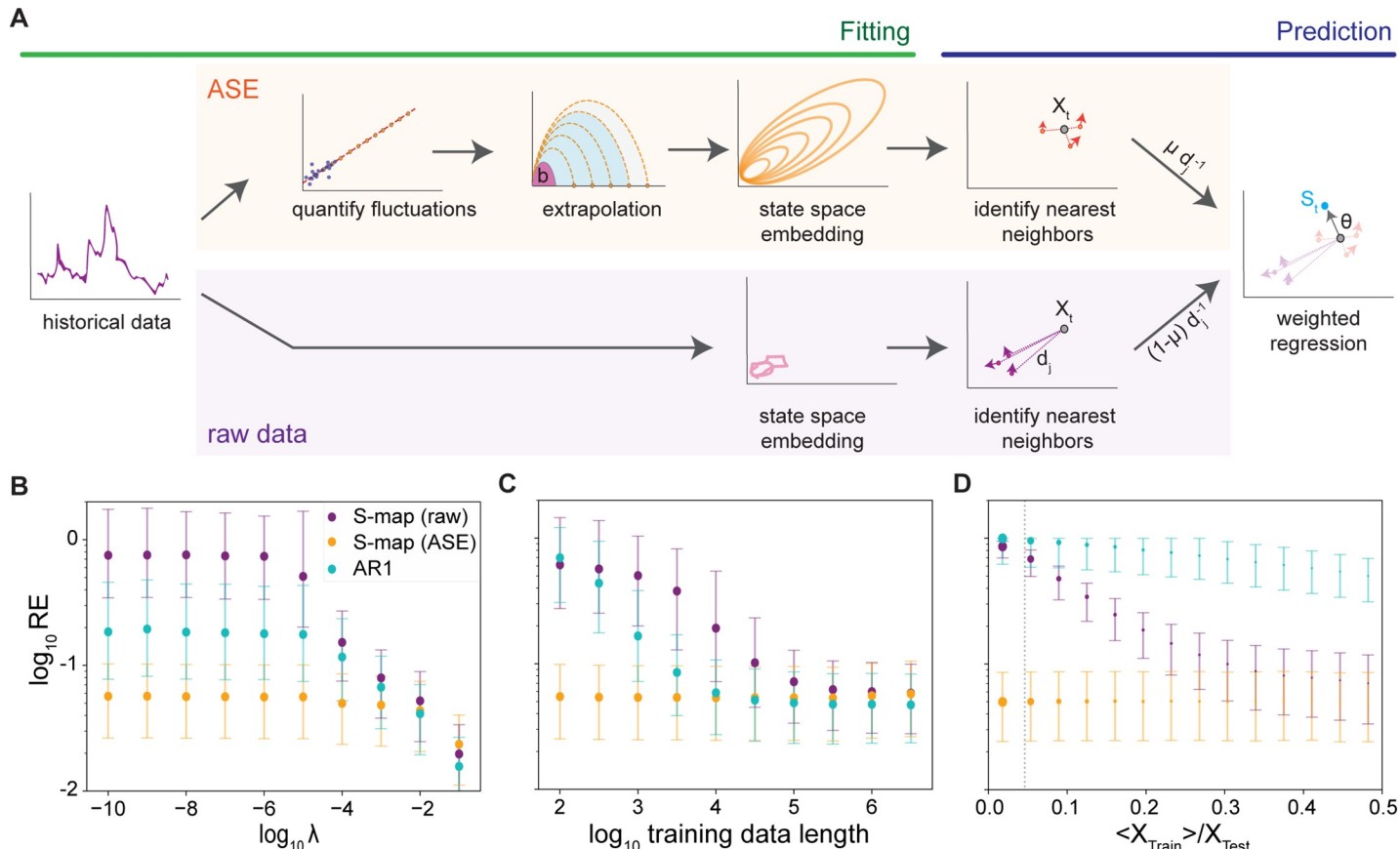

**Fig 2. ASE improves forecasting of fluctuations in simulated near-neutral dynamics. A,** Schematics showing ASE and S-map algorithms. Bottom: in standard S-map algorithm, 'fitting' is performed by embedding available historical data into a time-lagged state space. Predictions from a time point ($X_t$) are then made by identifying nearest neighbors in the embedded space and performing a linear regression weighted by their inverse distance ($\{d_j^{-1}\}$) from the current point to estimate local dynamics. The neighbors are inevitably far from the current point for unprecedented fluctuations. This estimate ($\theta$) is then used to predict future abundance ($S_t$). Top: ASE supplements the data in unsampled regimes. Avalanche events are segmented from historical data and used to fit the size-duration scaling $S = bT^\gamma$ (red line), where $\gamma = 3/2$ is set constant across communities. The unit duration ($b$) is used to generate ASE data (orange) using the scaling relationship and average avalanche trajectory (**Eq 2**). The ASE data are fed into S-map algorithm and integrated with the raw data for forecasting. **B,** Relative prediction error (RE) on dynamics with various degree of marginal stability using S-map with ASE only (orange), S-map with raw data (purple), and AR1 with raw data (cyan). For each condition 1,000 training sets were segmented randomly from a simulation (length = $10^8$) of **Eq 1** and used to predict 3,000 time points in a separate simulation. Training sets have a fixed length of 1,000 time steps. **C,** Relative prediction error vs. training data length. For each length, 500 training datasets were segmented randomly from a simulation (length = $10^8$) of **Eq 1** ($\lambda = 10^{-5}$). Each training dataset was used to predict 1,000 time points in a separate simulation. **D,** Relative error vs. training data average magnitude. The training sets have a fixed length of 100 time steps. The predictions were made on the largest 5% of values in the simulated testing data ($\lambda = 10^{-5}$). Dashed line: the median magnitude of training datasets, showing >50% of training data fall within the first bin. Symbol size scales with the number of data points in each bin. To accumulate statistics, 5,000 training sets were segmented and used to predict 5,000 points in a separately simulated dataset. In **B-D**, embedding dimension: m = 10, prediction time: P = 50. Symbols: median; error bars: interquartile values.

contributions weighted by their inverse distance ($d^{-1}$). This method suffers when no nearby neighbors are available and linearized dynamics must be extrapolated from other regimes. In these cases, nonlinear extrapolation by ASE should provide a better estimate of unsampled regimes (**Fig 2A**, prediction sector).

ASE is anticipated to offer three advantages: first, it relies on a single fit parameter from the data, reducing the required information content for forecasting; second, it provides a dense reconstruction of the state space that is difficult to sample in experimental time scales; third, it uses average trajectories for state reconstruction instead of raw trajectories, which inevitably contain sampling noise.

We ran several benchmarking tests on the data generated through the linear response model to quantify the value of ASE data in forecasting fluctuations. To characterize the dynamic regime in which ASE applies, we generated training data using various $\lambda$ and assessed the accuracy of forecasting time points in a separately simulated testing dataset using S-map with ASE data, S-map with raw data, and a first order autoregressive (AR1) method with raw data [37]. Though AR1 is a less general method than S-map, we include it here as a reference because it matches the linear assumptions of the simulated model. As expected, we found that using ASE data in S-map significantly reduces the prediction error compared to either method using raw data when $\lambda$ is small (**Fig 2B**). This is because as $\lambda$ becomes smaller, the probability of extreme, unsampled dynamics increases, resulting in poor extrapolation from raw data alone. In contrast, as deterministic dynamics dominate the system ($\lambda$ becomes larger), the differences between the methods diminish.

Next, we showed that ASE removes sensitivity to the availability of historical data. To do so, we ran each forecasting method using training datasets of various lengths. Predictions made by S-map with ASE data maintain almost constant small errors across all training data lengths, whereas shorter training data (sampling less of the fluctuation range) cause larger errors using S-map or AR1 with raw data (**Fig 2C**). This contrast persists to the limit where training data length approaches the forecasting time, at which point S-map with raw data is unable to execute. This feature of ASE is crucial in practice because there is often a lack of data in ecology [15].

To directly demonstrate that ASE improves performance through better extrapolation during unsampled events, we segmented data from the linear response model into training sets with constant length but varying fluctuation amplitude. We then compared the effect of training data amplitude when predicting the largest 5% amplitude time points using raw or ASE data (**Fig 2D**). We found that using ASE data in S-map enables accurate prediction of extreme events from training data that only contain fluctuations of small magnitudes. This is important as small fluctuations typically make up the majority of real ecological datasets.

Finally, we applied ASE to predict large rare events in the empirical data. To generalize beyond the regime close to marginal stability, we developed a hybrid version of S-map. In this method, raw and ASE data are each used to generate a separate reconstructed state space. Nearest neighbors are then selected from each reconstruction and weighted inversely with their distance from the current time point (d). To combine the two sets of nearest neighbors, we further weight terms from the raw data and ASE reconstructions by $1-\mu$ and $\mu$ respectively before performing the prediction step (**Fig 2A**). This method allows the ASE data to act as a prior condition when fitting local dynamics, i.e., when there is little historical data near the current time point, the prediction is dominated by the ASE data. The weighting parameter $\mu$ allows for tunable weighting of this prior condition and has been set at $\mu = 0.5$ throughout our analysis.

To assess forecasting accuracy, we segmented each empirical species dataset from the three ecosystems we study into short time series, each sampling a limited dynamic range. The short training set length allowed us to segment a larger number of independent training sets to evaluate statistical significance when comparing across methods. We used the 50% of segments with smallest magnitudes as training data to predict the 5% largest out-of-sample time points to show that our method is capable of better prediction of fluctuations unprecedented in historical data. ASE results in significantly reduced error across species and communities (**Fig 3**). Even for the small datasets of cyclopoids in plankton community and barnacles in the intertidal community that do not provide sufficient statistics for us to recover avalanche scaling laws, augmenting the historical data with ASE data still significantly improves forecasting accuracy.

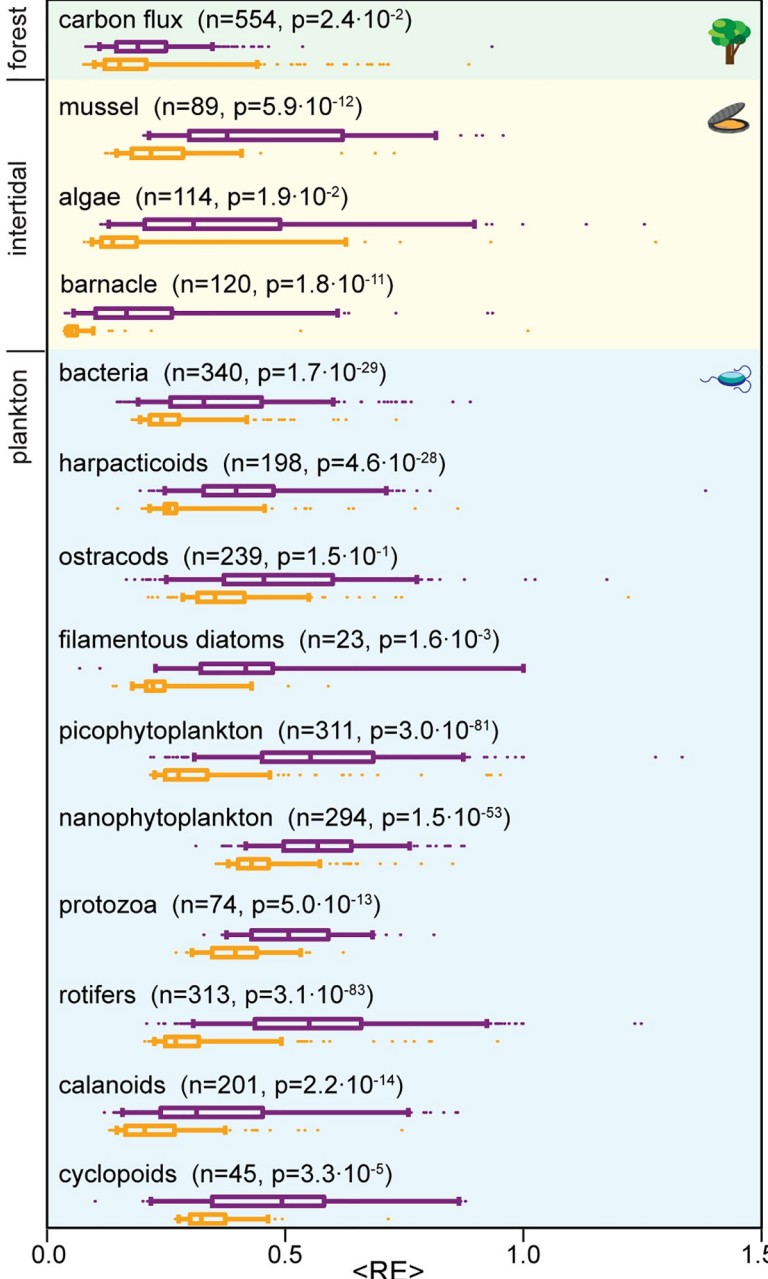

**Fig 3. Augmenting historical data with ASE improves forecasting in empirical data.** Comparison of S-map forecasts made from raw data (purple) and the hybrid method using both raw and ASE data (orange). To evaluate performance during unprecedented events, we segment empirical data into short segments of lengths specified below. We then use the 50% of segments with lowest average amplitude as training datasets to predict the 5% of time points with largest amplitude. The mean relative error across predictions, $\langle RE \rangle$, is computed for each of these segments and compared between the two methods. Data segment length: 10 time steps for plankton (equivalent of 1 month) and intertidal (equivalent of 10 months) data, 48 steps (equivalent of 1 d) for the forest data. Prediction time: 1 time step for plankton and intertidal data, 4 steps for forest data. Embedding dimension: m = 3 for all species. Bars: median; boxes: interquartile, whiskers: 5 and 95 percentile; dots: outliers. p-values are computed using a two-sided t-test to compare the mean errors in prediction with raw data only versus the hybrid method. Predictions are made for n training segments in each species, as specified in the figure.

## ASE enhances forecasting of chaotic dynamics

Does ASE also apply to ecosystems that do not follow near-neutral dynamics? To answer this question, we quantified the forecasting accuracy on barnacle abundance fluctuations in the mechanistic model derived from the intertidal community [3]. Using this model allows us to explicitly explore the effect of stochastic and chaotic forcing in forecasting accuracy. The barnacle abundance in this ecosystem can be strongly influenced by seasonal fluctuations. The strength of this effect was modeled through a tunable forcing parameter ($\phi$) known to drive the system towards chaos through a series of period-doubling bifurcations [3]. Additionally, the dynamics are influenced by a day-to-day stochastic temperature variation, characterized by a standard deviation $\zeta$.

As the system becomes more chaotic (larger $\phi$) the size distribution of species abundance fluctuations changes from a unimodal distribution, which is easy to sample with limited data, to increasingly broad distributions with complex shapes (**Fig 4A**). None of these size distributions follows a power law. However, while the fluctuation size-duration relation also deviates from a power-law scaling in the chaotic regime, it does mostly follow a monotonic trend (**Fig 4B**). The average fluctuation trajectories still exhibit approximately parabolic shape but are skewed significantly as the system enters the chaotic regime (**Fig 4C**). All these statistical irregularities suggest that it should be challenging for ASE to improve forecasting accuracy in chaotic ecosystems.

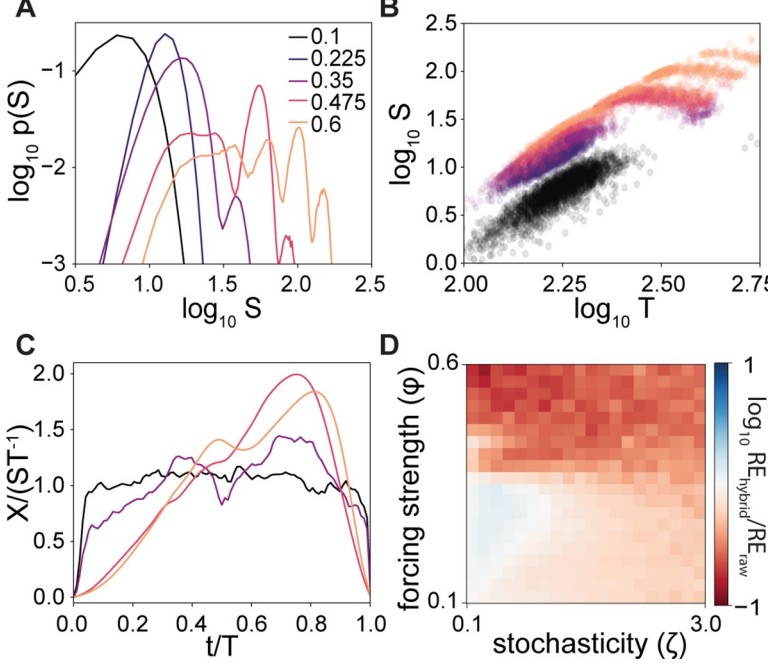

**Fig 4. ASE improves forecasting in chaotic regime. A,** Distribution of fluctuation sizes in barnacle abundance from the intertidal community model. As the system becomes more chaotic (larger $\phi$) the fluctuations go from an easily sampled unimodal distribution to broader, more complex shapes. Results are shown using data with the stochasticity term $\zeta = 1$. **B,** Scaling of fluctuation size vs. duration in barnacle abundance from the intertidal model. Colors same as in **A. C,** Average trajectories of barnacle fluctuations from the intertidal model. Colors same as in **A. D,** Comparison of relative prediction error using raw data and hybrid method. Training data length: 1 year of modeled data; prediction time 10 days. Embedding dimension: m = 3 for all conditions. The fold changes are averaged from 50,000 predictions at each condition, i.e., 500 time points randomly selected from separately simulated data were each predicted using 100 different training datasets.

We compared the forecasting accuracy of S-map using raw historical data alone and the hybrid method across a range of chaos and stochasticity. We found that as the system becomes more chaotic, historical data captures less of the full state space, and as an outcome, augmenting the training data with ASE data improves forecasting accuracy compared to the predictions using historical data alone. At the same time, as short-term stochasticity in the driving term increases, ASE reduces sensitivity to data idiosyncrasies and improves relative forecasting accuracy (**Fig 4D**). This is possible as ASE does not rely on specific assumptions about whether the stochastic driver of dynamics is due to chaotic microscale dynamics, intrinsic fluctuations, or environmental modulations. Taken together, these results demonstrate that ASE serves to improve forecasting in data-limited chaotic ecosystems.

In this analysis, we did not attempt to feed re-parameterized fluctuation size-duration relation or average fluctuation shape into ASE under each condition. This is because these statistical features are sensitive to conditions in the chaotic regime. Extracting them from real data requires large volumes of data and is typically impractical. Here, just as neutral ecology often serves as the null model for species interaction, we use ASE as a null predictive model, when the knowledge of species interactions is incomplete.

## Discussion

We study the scaling of ecological fluctuations in three independent long-term datasets. The scaling laws emerge from cumulative interactions within the communities and are consistent with near-neutral ecological dynamics [30]. The main advance of this work is a novel data extrapolation method (ASE) which leverages the emergent scaling phenomena to improve forecasting accuracy during rare events. ASE accomplishes this by extrapolating dynamics to historically unsampled regimes in a nonlinear state space. While prior work has focused on describing these fluctuations [21–27], our progress expands on and applies these results to gain predictive power.

Compared to existing data-intensive ecological forecasting algorithms [10,13,15], ASE achieves its superior performance by relying on only a single fit parameter, significantly reducing sensitivity to the quality and volume of the training data. In particular, by utilizing the average trajectories it reduces the risk of overfitting to the idiosyncrasies of individual fluctuations. These features are essential to enable accurate forecasting in understudied communities, where extensive historical information does not exist.

The augmented data produced by ASE can be directly used in S-map as well as other locally-weighted methods for state space reconstruction. Alternatively, for forecasts based on mechanistic models, ASE data can be used to define null-model based priors for fit parameters. Here, the goal of ASE is not to recover the detailed interactions of a community that give rise to the observed dynamics but to capture the generic statistical regularity that can be used to inform predictions.

We anticipate ASE to be useful even in systems where significant prior data is available. First, the observed power-law distribution in fluctuations suggests unprecedented events will occur regardless of the length of observation time. This illustrates an intrinsic challenge for ecological forecasting: data never appears sufficient to capture the most impactful events. To address this, ASE takes advantage of the "scale-free" property of these fluctuations to extract information contained in frequent small-amplitude events and extrapolate it to historically unobserved regimes.

Second, it is well-documented that ecosystems can suddenly shift to new equilibria due to species invasion or long-term changes in environmental conditions such as climate change [16,38,39]. Species interactions, and thereby fit model parameters, can be significantly altered

in these scenarios, requiring retraining of the model with newly acquired data [40]. In these cases, ASE can more rapidly provide useful predictions for the new community assembly while data around the new equilibria is still limited.

One obvious potential limitation of the ASE method is that not all ecosystems follow near-neutral dynamics. As a result, fluctuations in those systems must deviate from the avalanche scaling behaviors, which might compromise the forecasting accuracy of ASE. However, we have demonstrated that augmenting the historical data with ASE data can improve forecasting accuracy even in a chaotic regime. This implies that ASE provides a better extrapolation of unobserved fluctuations than the standard S-map method. ASE also ignores the fact the fluctuations in real ecosystems must be constrained by the physical limits of carrying capacity, because it is difficult to estimate these values from limited historical data. This finite size effect can introduce truncations in avalanche scalings and cause errors in ASE, especially in systems with small carrying capacities. While it exits as a theoretical possibility, we have not observed this effect in our analyses of empirical data that were collected from experimental and natural ecosystems.

While this work focuses on the application of ASE to single species and ignores spatial structures, the technique can be extended to analyze multispecies interactions and spatial correlations. Fluctuations can be fit and predicted along the principle components of multidimensional datasets. These high-order complications can potentially alter the avalanche statistics [17], therefore, ASE should again serve as a null model–any deviations would then require fitting system specific community models from this prior.

## Methods

### Data

This work studies three experimental and natural ecological communities, which were selected based on previously reported "edge-of-chaos" nonlinearity signatures, relatively high frequency measurements, and long term monitoring—enabling us to identify a large number (>100) of avalanche events from each dataset.

*Plankton*: Data for this community come from a ~8 year study of a plankton community isolated from the Baltic Sea and cultured in a laboratory mesocosm under constant external conditions [4,28]. Species abundances were measured twice weekly as a fresh weight biomass. Trophic groups were defined in Ref. [4] as producers (picophytoplankton, nanophytoplankton, filamentous diatoms), herbivores (protozoa, rotifers, calanoids), and detritivores (bacteria, ostracods, harpacticoids). Predatory zooplankton (cyclopoids) were excluded from scaling analyses due to the reduced quantity of data at this trophic level. The time series data were accessed from Ref. [28] without further processing. To obtain sufficient events to characterize the avalanche size distributions in the plankton community, we pooled segmented fluctuations from all species at each trophic level.

*Forrest*: Data for this community come from the Ameriflux site US-HA1 and are accessible at http://dx.doi.org/10.17190/AMF/1246059. This is a deciduous broadleaf forest with data spanning from 1991 to present. Net carbon flux was recorded made in 30 minute intervals at a height of 30 m [29]. As in previous work analyzing this time series [2], we linearly detrended the data prior to our analysis in order to isolate stochastic avalanche fluctuations from long term shifts in forest productivity.

*Intertidal*: Data for this community come from a 20 year study on the interactions of barnacles, algae, and mussels on the coast of New Zealand [3]. Abundances were measured as a percent of rock coverage on a monthly basis.

Additionally, the original study developed and fit a mechanistic ODE model of this community. In this model, barnacles colonize bare rock. Algae attach and grow on bare barnacles with no effect on the barnacles. Mussels, which cannot colonize bare rock alone, adhere on top of and smother the previous two species, increasing the detachment rate of them and thereby driving the system back to bare rock. Additionally, the detachment rate of all species is influenced by temperature. This variation is controlled by a seasonal oscillation, scaled by a parameter, $\phi$, and a day-to-day stochastic variation sampled from a normal distribution with mean zero and standard deviation $\zeta$.

For the avalanche scaling analysis (**Fig 1**), we used the model with all parameters specified in the original work to generate additional time series. To focus on species fluctuations due to marginal stability, we set the seasonal variations stationary ($\phi = 0$) and stochasticity $\zeta = 2$. Unlike algae and mussels, barnacle abundance in this model is maintained at a rather steady high level under this condition. We therefore excluded barnacle from this part of the analysis. For analysis on chaotic dynamics (**Fig 4**), we generated time series data of barnacle abundance using non-zero $\phi$. At small $\phi$, the system undergoes yearly oscillations with the seasonal forcing. As $\phi$ increases, the system is driven to chaos through a series of period-doubling bifurcations [3]. Analyses of algae or mussels led to qualitatively similar conclusions, which are not discussed in the text for brevity.

## Avalanche analysis

We defined the reference baseline level as the average measured value through a time series. The exact reference value, however, does not affect the scaling of avalanche events (**S1 Fig**). Therefore, in the plankton community we lowered the threshold to 10% of the average to include more small fluctuations, which significantly improved the confidence of the fit exponents.

Avalanche events were identified as deviations of the tracked species abundance (X) above the baseline. Duration (T) was determined by the time between exceeding and falling below this threshold. Avalanche size (S) was calculated as the Reimann sum of the area between the threshold and measured species abundance through the duration of an event.

To plot the probability distribution of avalanche durations and sizes (**Fig 1B and 1C**) and the average size vs. duration relation (**Fig 1D**), we placed events in logarithmically scaled bins to improve sampling of low frequency regions. To determine the relative frequency of events with different durations and sizes, we normalized the number of events in each bin by either the number of time points contained within when the data are discrete or the bin width when the data are continuous.

Power law fitting of the frequency distributions were verified by comparing the Akaike information criterion (AIC) for analytically derived maximum likelihood estimates of power law and exponential scaling (**Table 1**). This approach is independent of any binning process, providing a less biased measurement of power law behavior [32]. Scaling exponent of the average avalanche size was determined by fitting a linear regression to bin-averaged data values in log-log scale (**S1 Table**).

To calculate the average avalanche trajectory, individual events were isolated and normalized to unit duration ($t' = t/T$) and unit size ($X' = X/ST^{-1}$, resulting in $S' = \int_0^1 X'(t')dt' = 1$). The average curve was then calculated by averaging the normalized abundance at each normalized time point. Because the data from differently sized events were unevenly sampled after temporal normalization, we used interpolated values of each trajectory to achieve uniform time sampling for averaging. To improve convergence in limited empirical data, we lumped fluctuations from all species within each community for this calculation.

## Linear response model

While the identified scaling laws are consistent with previous theoretical predictions [20,30], forecasting critical events requires a detailed average fluctuation trajectory, which is not provided in the previous theoretical work. To generate this trajectory, we used a linear response model at marginal stability, an inherent regime of high-dimensional complex ecosystems as suggested by classic theoretical analyses [34,36].

Consider an ecosystem of many interacting species. Dynamics can be written generally as $\frac{d\vec{s}}{dt} = A\vec{s}$, where $\vec{s}$ is a vector of species abundance and A is the interaction matrix. In general, $\vec{s}$ can also contain abiotic components of the system (e.g., $CO_2$ flux) and A is a non-linear function of the system state. If there exists a fixed point such that $\frac{d\vec{s}}{dt}\big|_{\vec{s}_s} = 0$, its stability can be evaluated by determining the eigenvalues of $A\big|_{\vec{s}_s}$, with each eigenvalue $\lambda_i$ defining the linear response along the direction of the corresponding eigenvector $\vec{x}_i$. If $A\big|_{\vec{s}_s}$ is negative definite ($\lambda_{max} < 0$), the fixed point is stable and perturbations decay to equilibrium.

Previous work analyzing the properties of $A\big|_{\vec{s}_s}$ has identified two key features: first, as the dimension increases ($\vec{s}$ contains many species/components) the probability of a fixed point being stable drops dramatically [36], and second, almost all of the few stable fixed points in this limit are marginally stable, i.e., $\lambda_{max} \to 0$ [34,35]. This means that in a high-dimensional ecosystem with many weak interactions, we should expect to observe dynamics near marginally stable fixed points. In addition, transient marginal stability can also arise in ecosystems as the dominant eigenvalue approaches zero around tipping points [16,39]. However, the specific generative mechanism is irrelevant to the determination of scaling laws.

To simulate dynamics data at marginal stability, we introduce a multiplicative noise representing stochasticity in birth, death, or other cumulative rates to evaluate local dynamics around a fixed point, as given by:

$$x_{t+1} = \eta_t(x_t + \lambda(x_s - x_t)) \tag{1}$$

where $\lambda$ and $x$ are the dominant eigenvalue and position along the dominant eigenvector respectively. $x_s$ is the equilibrium value of $x$, and $\eta_t \sim N(1, \varepsilon)$. The projection of $x$ onto any species component is a linear transformation, and as a result, scaling laws in the dynamics of $x$ should be observed in the time traces of all non-orthogonal measurements of the community, including species abundances and metabolic rates (e.g., the carbon flux of the forest community).

For numerical simulation, **Eq 1** was integrated using an explicit Euler integration function. Simulations were run with $\eta_t$ independently sampled at each time step from a Gaussian distribution with mean 1 and standard deviation 0.01. The simulations produce avalanche dynamics across an increasingly wider dynamic range as $\lambda \to 0$ (**S2 Fig**), with exponents in quantitative agreement with the empirical data (grey lines in **Fig 1B–1E**).

In this model, we ignore the potential environmental influences. However, we note that the plankton community studied in Ref. (28) was grown in a closed laboratory system under largely constant conditions, suggesting that exogenous fluctuations are not necessary for the observed power law statistics. Additionally, the stochastic noise in the forcing term (i.e., temperature) in the intertidal model is Gaussian distributed, indicating that the power law statistics do not directly correlate with the external driving force.

## Avalanche scaling extrapolation (ASE)

ASE first defines average avalanche events from historical data and fits their size and duration to the power-law scaling, $\langle S_T \rangle = bT^\gamma$, in log-log space (**Fig 2A**). The scaling exponent (slope, $\gamma$)

is assumed to be conserved across systems with near-neutral dynamics, allowing ASE to only fit for the y-intercept (*b*), which is the size of a unit duration avalanche. The normalized avalanche trajectory $Y_n(t')$ is generated through quadratic interpolation of the converged curve from the linear response model as shown in **Fig 1E**. It is then scaled according to the size-duration relation (**Fig 1D**). This produces a set of avalanche trajectories $\{Y_T(t)\}$ spanning all time-scales of interest, where:

$$Y_T(t') = Y_n(t') * \langle Y_T \rangle = Y_n(t') * S_T T^{-1} = Y_n(t')[bT^{\gamma-1}] \qquad [2]$$

and $t' = t/T$. These trajectories effectively sample the entire state space outside the historical data and are appended to create an augmented training dataset.

## Forecasting method

*S-map*: Raw historical and ASE data are fed into the single species S-map prediction algorithm [13]. The algorithm is based on Taken's theorem, which states that the information about a D-dimensional attractor can be captured in an m-dimensional embedding of the time series of a single state of the system, if $m \geq 2D+1$ [41]. Intuitively, S-map estimates the local linearized dynamics based on a distance-weighted regression of neighboring data points in the embedded space. The local weighting allows this method to account for dynamics that vary non-linearly across the state space without an explicit model of the system. Because S-map performs an averaging of local states, it is well suited for using the trajectories generated by ASE which are averaged dynamics for unobserved fluctuations across various time and length scales. Theoretical justification of S-map even in the presence of internal or exogenous noise is available in Ref. [9,42].

In the S-map algorithm, for each time point in the training data, we define the embedding vector $X_{j,t} = \{x_t, x_{t-\tau}, \ldots, x_{t-(m-1)\tau}\}$ where m is the embedding dimension and $\tau$ is the interval time. To make prediction of $x_{i+P}$, where P is the prediction time, we define the embedding vector $X_i$. We then identify the k-nearest neighbors to $X_i$ in the training data $\{X_{j,h}|h \in \{1,..k\}\}$ based on the 2-norm. These nearest neighbors are appended into the [$m \times k$] matrix $A_i$ and their corresponding accurate predictions into a *k* length vector $S_i$. We then solve the weighted least square regression $WS_i = \theta(A_i W)$ for the mapping function $\theta$, where W is a [$k \times k$] matrix used to weight each nearest neighbor $(X_{j,h})$ by $\|X_i - X_{j,h}\|^{-q}$. The prediction for the current time point can then be made as $X_{i+P} = X_i \cdot \theta$.

Our hybrid method modifies the S-map algorithm by combining nearest neighbors from both raw and ASE data for the weighted linear regression. Specifically, each dataset is embedded separately. Then, *k* and $k_{ASE}$ nearest neighbors are selected from the raw and ASE embeddings respectively and weighted by their respective inverse distance. Each neighbor from the raw and ASE datasets are then weighted by an additional factor of $\frac{(1-\mu)}{k}$ and $\frac{\mu}{k_{ASE}}$ respectively, where $\mu \in [0,1]$. The weighting parameter $\mu$ allows us to control the strength of the prior given by ASE dynamics. The nearest neighbors are then used for weighted linear regression.

$q = 1$, $k = 2$ were used for all forecasting analyses in this work and were chosen to optimize performance of predictions on the marginal stability model using the raw data alone. A default value of $\mu = 0.5$ was used for all predictions based on both raw and ASE data. Due to differences in sampling frequency and data availability, *m*, $\tau$, and *P* were varied between species as specified in the figure captions. Because forecasting requires evenly spaced time points, we ran predictions for plankton and intertidal data on interpolated values provided in the original studies [3,28].

*AR1*: AR1 makes predictions using a linear regression, $x_{t+P} = \theta_0 + \theta_1 x_t$. Unlike in S-map where reconstructions are based on local dynamics, in AR1 all timepoints in historical data are

used to estimate $\theta$ using least squares regression. Note that this is fully equivalent to the specific case of S-map where $m = 1$, $p = 0$, and $k = n$, where $n$ is the total number of points in historical data.

## Supporting information

**S1 Fig. Avalanche scaling laws are independent of reference level. A-B**, Probability distribution scaling in herbivorous plankton (**A**) and linear response model (**B**) is independent of reference abundance. The references are in the unit of mean abundance. Shifted power-law cutoff (arrow) is due to higher reference that truncates the avalanche duration. **C-D**, Size-duration scaling exponent independent of threshold abundance. Data in **C** are noisier at higher references due to the loss of events and limited statistics. Symbol color: same as **A** and **B**.
(TIF)

**S2 Fig. Marginal stability produces the universal scaling law. A**, Perturbations of ecosystem state (green dot) from a fixed point (black or grey filled symbol) respond along the eigenvectors (red and blue arrows). Top: all eigenvalues are negative. The system is stable and follows a deterministic exponential decay to equilibrium (green arrow). Bottom: one or more eigenvalues approaches zero, creating marginal stability. The deterministic linear response becomes weak relative to stochastic terms (pink arrow), which then drives the system fluctuations along the corresponding eigenvector. **B**, Probability distribution of avalanche durations in the linear response model (**Eq 1**) approaches a power law at marginal stability as $\lambda \rightarrow 0$. Line: reference slope of 3/2. Simulation run with $\eta \sim N(0, 0.01)$.
(TIF)

**S1 Table. Statistics of average avalanche size scaling.** Bin average avalanche sizes of individual species from **Fig 1D** were fit to the equation using linear regression in log-log scale. s.e.: standard error.
(PDF)

## Acknowledgments

We thank K. Chen and S. Granick for critical reading of the manuscript.

## Author Contributions

**Conceptualization:** Samuel R. Bray, Bo Wang.

**Data curation:** Samuel R. Bray.

**Formal analysis:** Samuel R. Bray.

**Funding acquisition:** Bo Wang.

**Investigation:** Samuel R. Bray.

**Methodology:** Samuel R. Bray.

**Project administration:** Bo Wang.

**Software:** Samuel R. Bray.

**Supervision:** Bo Wang.

**Validation:** Samuel R. Bray.

**Visualization:** Samuel R. Bray.

**Writing – original draft:** Samuel R. Bray, Bo Wang.

**Writing – review & editing:** Samuel R. Bray, Bo Wang.

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
