## [Decision Letter · Decision Letter 0]

29 Jan 2020

Dear Wang,

Thank you very much for submitting your manuscript "Forecasting unprecedented ecological fluctuations" for consideration at PLOS Computational Biology.

As with all papers reviewed by the journal, your manuscript was reviewed by members of the editorial board and by several independent reviewers. In light of the reviews (below this email), we would like to invite the resubmission of a significantly-revised version that takes into account the reviewers' comments.

We cannot make any decision about publication until we have seen the revised manuscript and your response to the reviewers' comments. Your revised manuscript is also likely to be sent to reviewers for further evaluation.

Sincerely,

Jacopo Grilli

Associate Editor

PLOS Computational Biology

Stefano Allesina

Deputy Editor

PLOS Computational Biology

Reviewer's Responses to Questions

**Comments to the Authors:**

Reviewer #1: In this manuscript the authors analyze the distribution of population peaks in some data sets. They call the peaks “avalanches” which is confusing as they suggest that the power law is not due to self-organized criticality (suggesting a build-up of tension), but simply due to random walk. Similar systems are earthquakes and forest fires. They fit a power law to the population sizes and present a way to predict extreme events based on extrapolation.

Although the fitted power laws are strikingly similar among different data sets, I am still not convinced of the suggestion that we can use extrapolated power laws to predict extreme events. That is because of several reasons.

1. The theory that all ecosystems are neutrally stable and thus exhibiting random walk is not sell-established in ecology as far as I know. I don’t see how this can explain the tree productivity data.

2. I am not convinced by the tests of the extrapolation tests. If I understand correctly this testing is done in two ways: (1) linear marginally stable model (2) using the low values of the data to predict the peaks. (1) Why do the authors use a LINEAR model that is neutrally stable to simulate species. It doesn’t make sense as the prediction method (S-map) is based on a non-linear model. I could not find the reason why all high dimensional systems should be marginally stable (I could not find that in the cited paper of Stone). That would mean they are (2) It seems obvious that if you predict extreme values from low values that the prediction becomes better if you allow extrapolation. It doesn’t seem a fair test (though I might misunderstood the methods, see next point).

3. The figures and method section are very hard to understand. The symbols in the figure axes are not explained (for instance how was Figure 1E constructed and what is on the axes and what does this curve tell me?). Also in the text, not all symbols are explained (T. The structure of the paper is very strange. An important part of the methods are not explained in the method section but in the last part of the result section. It was completely unclear to me what the role is of the linear model when I read the method section. I was searching for details on the ASE method, which was in the result section. I knew the S-map method already, but that would probably also difficult to understand for ecologists that are not familiar with that. Some descriptions in the methods are very condensed and I found it hard to understand what was exactly done. For instance "The normalized avalanche trajectory (′), generated through quadratic interpolation of the converged curve from the linear response model as shown in Fig. 1E, is then scaled according to the size-duration relation Fig. 1D" What makes it extra difficult is that the symbols in the figures are not explained in the legends (later I saw the small figure

4. It was not clear to me how the trajectories were averaged. Why can an average trajectory be used to predict events? The S-map method seems especially suited to predict non-standard trajectories, as it is assumes a response to be state dependent. Moreover I don’t think S-map can predict random walk data sets (as suggested by the marginal stability) as they are fully driven by stochasticity.

5. I think there is a physical maximum to the size of the population/productivity peaks. This should be due to resources. The extrapolation method does not seem to account for that, but could predict unrealistically high peaks in biomass/productivity.

6. Can chaotic dynamics also explain the power law that was found? At least in the used plankton data set, chaotic dynamics were demonstrated. It would be good to check for power law distributions in models that generate such chaotic dynamics (for a model see for instance Dakos et al. 2009 Proc. R. Soc. B (2009) 276, 2871–2880).

Reviewer #2: These authors found that several different ecological datasets showed fluctuations that obeyed scale-free statistics, and that could be modeled using statistical models that were developed for avalanches. Using this realization, one can predict the frequency of extreme events, including those that are outside of any observed events, much more accurately than is possible if one does not account for it.

I found the paper to be well-written and interesting, and the research performed well. At least in retrospect, these results don’t seem tremendously profound because it seems reasonable that one could use statistics of past fluctuations to predict future fluctuations, including large ones. I expect that this approach is common in flood forecasting (e.g. the 100-year flood) and in earthquake forecasting. However, I am not a theoretical ecologist, so this may be a new insight to ecology research. In any case, it’s an important one.

As a critique of the work, I’d like to know more about the limits of this approach, showing where it does and doesn’t apply. As part of this, the data sets were chosen carefully to be at the edge-of-chaos and of high quality, with frequent samples taken over a long time period. Would the theory still apply if the ecological system were not at the edge of chaos (e.g. the barnacles in the intertidal data set)? Also, can useful things be said about the data sets that are of lower quality (e.g. the cyclopoids in the plankton data set), or at least predictions made for them?

There is no mention as to the cause of the fluctuations here, leading me to initially assume that they arose from intrinsic ecological chaotic dynamics. However, I realized afterward that the primary cause might simply be the weather or other environmental perturbations, especially for the forest and intertidal data sets. Thus, is this fluctuation analysis really an investigation of the statistics of ecological fluctuations, or of environmental fluctuations? If it’s the latter, then it’s still useful for applied work, but it changes the interpretation substantially. Also, it leads to the question of whether weather obeys avalanche statistics, and if that’s the primary driving force for these ecological fluctuations and perhaps others as well.

In the Linear Response model section, did you use random matrices or non-random matrices? My understanding is that May’s work found that large random matrices had few stable fixed points, and marginal stability for those that were stable, as the text describes here. However, I think more recent work has largely refuted those results because May used random matrices whereas real ecological systems are very non-random, and many of those turn out to be much more stable than originally believed. If this understanding is correct, then this might need to be addressed in your work.

I didn’t really understand the Forecasting method section. Perhaps a figure would help for the explanation, or at least a more detailed explanation of the background material.

Reviewed by Steve Andrews

**Have all data underlying the figures and results presented in the manuscript been provided?**

Reviewer #1: Yes

Reviewer #2: Yes

PLOS authors have the option to publish the peer review history of their article (what does this mean?). If published, this will include your full peer review and any attached files.

Reviewer #1: No

Reviewer #2: Yes: Steven S. Andrews
---

## [Decision Letter · Decision Letter 1]

20 Apr 2020

Dear Wang,

Thank you very much for submitting your manuscript "Forecasting unprecedented ecological fluctuations" for consideration at PLOS Computational Biology. As with all papers reviewed by the journal, your manuscript was reviewed by members of the editorial board and by several independent reviewers. The reviewers appreciated the attention to an important topic. Based on the reviews, we are likely to accept this manuscript for publication, providing that you modify the manuscript according to the review recommendations.

Sincerely,

Jacopo Grilli

Associate Editor

PLOS Computational Biology

Stefano Allesina

Deputy Editor

PLOS Computational Biology

[LINK]

Reviewer's Responses to Questions

**Comments to the Authors:**

Reviewer #1: In this revision the manuscript has substantially improved. Still I have problems with understanding this work, maybe because of my limited knowledge of this kind of methods.

As I understand the linear model, it seems that it is an autoregressive model, that actually seems to produce Brownian motion if lambda goes to zero. I do not understand that such model is so good to predict (for instance lambda 1E-10 in figure 2D should be very close to Brownian motion). Maybe it is because the S-map method on the raw data is performing very badly as it is not suited for this? I would be interested in seeing how another basic prediction method would perform, like tomorrow will be the same as today, or an AR1 model? Is the ASE method also much better than this?

Some parts of the methods are really hard for me to follow. What is exactly the role of the standardized avalange histories in the prediction. How are these included in the S-map method?

The division of the data in a training set and a prediction set is not easy to understand. I only found some information in the legend of Figure 3. Why is the training data set length so short (only 10 points)? I do not understand the last sentence: “2n statistically independent data sets were segmented .... pvalues are two sided t tests?” What is 2n? 50% are used as training set? (but the length was only 10 points?). What is tested with the t-test?

If these points can be explained I think this manuscript can be published.

Reviewer #2: The authors have satisfactorily addressed all of my concerns.

**Have all data underlying the figures and results presented in the manuscript been provided?**

Reviewer #1: Yes

Reviewer #2: Yes

PLOS authors have the option to publish the peer review history of their article (what does this mean?). If published, this will include your full peer review and any attached files.

Reviewer #1: No

Reviewer #2: Yes: Steve Andrews
---

## [Editor Report · Decision Letter 2]

5 Jun 2020

Dear Wang,

We are pleased to inform you that your manuscript 'Forecasting unprecedented ecological fluctuations' has been provisionally accepted for publication in PLOS Computational Biology.

Best regards,

Jacopo Grilli

Associate Editor

PLOS Computational Biology

Stefano Allesina

Deputy Editor

PLOS Computational Biology

---

## [Editor Report · Acceptance letter]

22 Jun 2020

PCOMPBIOL-D-19-02185R2 

Forecasting unprecedented ecological fluctuations

Dear Dr Wang,

I am pleased to inform you that your manuscript has been formally accepted for publication in PLOS Computational Biology. Your manuscript is now with our production department and you will be notified of the publication date in due course.

With kind regards,

Sarah Hammond
